# Indian Medicinal Plant-Derived Phytochemicals as Potential Antidotes for Snakebite: A Pharmacoinformatic Study of Atrolysin Inhibitors

**DOI:** 10.3390/ijms252312675

**Published:** 2024-11-26

**Authors:** Deva Asirvatham Ravi, Du Hyeon Hwang, Ramachandran Loganathan Mohan Prakash, Changkeun Kang, Euikyung Kim

**Affiliations:** 1College of Veterinary Medicine, Gyeongsang National University, Jinju 52828, Republic of Korea; devabiochem@gnu.ac.kr (D.A.R.); pooh9922@gnu.ac.kr (D.H.H.); mohanprakash111@gmail.com (R.L.M.P.); ckkang@gnu.ac.kr (C.K.); 2Institute of Animal Medicine, Gyeongsang National University, Jinju 52828, Republic of Korea

**Keywords:** Atrolysin, snake venom, metalloproteinase, molecular dynamics, MMPBSA, density functional theory (DFT)

## Abstract

Snakebite envenoming is a significant health threat, particularly in tropical regions, causing substantial morbidity and mortality. Traditional treatments, including antivenom therapy, have limitations and associated risks. This research aims to discover novel phytochemical antidotes for snakebites, specifically targeting the western diamondback rattlesnake (*Crotalus atrox*) venom metalloproteinase Atrolysin. Utilizing pharmacoinformatic techniques such as molecular docking, high-throughput ligand screening, pharmacophore mapping, pharmacokinetic profiling, and molecular dynamics (MD) simulations, we analyzed phytochemicals from the Indian Medicinal Plants, Phytochemistry And Therapeutics (IMPPAT) database alongside well-known nine metalloproteinase inhibitors from the PubChem database. From an initial set of 17,967 compounds, 4708 unique compounds were identified for further study. These compounds were evaluated based on drug likeness, molecular descriptors, ADME properties, and toxicity profiles. Binding site predictions and molecular docking identified key interacting residues and binding energies, highlighting several promising compounds. Density functional theory (DFT) analysis provided insights into these compounds’ electronic properties and stability. MD simulations assessed the dynamic stability of protein-ligand complexes using parameters such as RMSD, RMSF, the radius of gyration, and hydrogen bond interactions. This study identified top candidates, including CID5291, IMPHY001495, IMPHY014737, IMPHY008983, IMPHY008176, and IMPHY003833, based on their favorable binding energies, interaction forces, and structural stability. These findings suggest that the selected phytochemicals have the potential to serve as effective alternatives to traditional antivenom treatments, offering a promising avenue for further research and development in snakebite management.

## 1. Introduction

Snakebite envenoming is an important but largely neglected health threat that impacts millions of individuals living in tropical regions across the world [1,2]. Annually, an astounding 5.4 million snake bites occur worldwide, resulting in 1.8–2.7 million instances of envenomation and 81,000–138,000 deaths [3]. According to the World Health Organization, an estimated 400,000 people suffer permanent deformities due to snake envenomation each year. These include limited mobility, chronic scarring, blindness, and even limb amputation, alongside unreported psychological trauma. Snakebite envenoming kills an estimated 20,000–32,000 people per year in Sub-Saharan Africa. However, these statistics may be conservative, as many sufferers prefer folk cures over medical care [4,5,6,7].

The western diamondback rattlesnake, or *Crotalus atrox*, is a dangerous pit viper belonging to the Viperidae family, which is a subfamily of Crotaline. It is known for its potent venom composition, which comprises an array of toxins [8]. This venomous secretion is a complex amalgamation of proteins, enzymes, peptides, and other bioactive molecules. The venom includes zinc metalloproteinases that disrupt the endothelium, leading to local and systemic hemorrhage, hypotensive shock, intravascular clotting, edema, pain, and necrosis [9,10,11,12].

Though scientific and technological innovations continue to progress rapidly, immunotherapy remains the primary approach for treating snakebite envenoming. Current antivenoms are essential for preventing fatalities, yet their limitations have encouraged researchers to explore alternative neutralizing agents that could enhance or work in conjunction with conventional antivenom therapies [13]. Phytochemicals can be used as they are found in nature or undergo derivatization before being included in drug formulations. In developed countries such as the United States, plant-derived drugs make up to 25% of the total drugs used in therapeutics, while in rapidly developing nations like China and India, this rises to as high as 80% [14]. Snakebite management often involves utilizing plant-derived therapeutics as a complementary treatment alongside antivenom administration [15,16]. Plant extracts are rich in various phytochemicals, some of which can neutralize venom toxicity. While plant extracts have a limited level of effectiveness in neutralizing toxicity, purified compounds can have a stronger effect [17]. Despite the widespread use of plant-based medicines for treating snakebite victims, there is still a lack of identified molecules capable of completely neutralizing snake venom toxicity [18,19,20]. Computational screening for identifying drug candidates offers a promising and cost-effective approach that is integral to the drug discovery and development process [21,22]. By leveraging computational chemistry, medicinal chemists can address a wide range of theoretical aspects involved in modern drug discovery. With advancements in computational tools driven by the enhanced capabilities of computers, supercomputers, and parallel computing, it is now feasible to manage complex tasks within the drug discovery pipeline [23,24,25]. The primary goal of this research is to discover novel antidotes for snake bites, particularly those caused by *Crotalus atrox,* Snake Venom Metalloproteinase (SVMP), and Atrolysin. The pharmacoinformatic techniques used in this work include molecular docking, high-throughput ligand screening, pharmacophore mapping, pharmacokinetic profiling, and molecular dynamics (MD) simulation analysis. This in silico investigation aims to gain a comprehensive understanding of the selected phytochemicals and their mechanisms for neutralizing the *Crotalus atrox* toxin, Atrolysin. The findings may also pave the way for further exploration into the potential use of these bioactive phytochemicals as alternatives to antivenom treatment, thereby mitigating the risk of adverse and life-threatening side effects associated with traditional treatments.

## 2. Results

### 2.1. Bioactive Molecules and Medicinal Plants Identified in the Study

The Indian Medicinal Plants, Phytochemistry And Therapeutics (IMPPAT) database was used in our study to identify phytochemicals for treating snake bites. Our initial exploration included 738 entries from various plant parts, such as bark, flowers, fruits, leaves, seeds, roots, and stems. After removing duplicates, we focused on 408 unique plant species and retrieved 17,967 compounds associated with these species from the IMPPAT database. We identified 4708 unique compounds of interest by eliminating duplicate compounds, as detailed in Appendix A. The ClassyFire SuperClass of phytochemicals is shown in Figure 1. 

### 2.2. Ligand-Based Screening and Pharmacokinetics Profiling for the Discovery of Promising Inhibitors of Snake Venom Metalloproteinases

Fundamental descriptors were acquired and summarized to understand the molecular properties of IMPAAT database compounds (Appendix A). Figure 2 illustrates the outcomes of ligand-based screening for IMPAAT database compounds, showcasing toxicity indexes, drug-likeness scores, and the distribution of physicochemical properties.

#### 2.2.1. Analysis of Molecular Descriptors and Pharmacokinetic Properties of Natural Compound

Figure 2a–f illustrates the molecular descriptors of natural compounds based on the reference values from Lipinski’s and Veber’s guidelines, represented by a red star. Among the 4708 natural compounds analyzed, a significant portion demonstrated favorable properties: 83.88% had a molecular weight of ≤500 g/mol, 78.54% exhibited acceptable hydrophobicity scores, 85.85% possessed a topological polar surface area of ≤150 Å^2^, and 88.7% had 10 or fewer rotatable bonds. Interestingly, only a small percentage of compounds surpassed the limits for the number of hydrogen-bond acceptors (88.14%) and donors (88.18%). This observation suggests that most of the analyzed natural compounds are likely to be absorbed into systemic circulation, resembling orally bioavailable drug candidates. Furthermore, the predicted Absorption, Distribution, Metabolism, and Excretion (ADME) properties of these compounds predominantly fell within favorable ranges, consistent with Lipinski’s and Veber’s rules. The compounds were subjected to various toxicity and stability assays to further define their profiles. In the cytotoxicity assay, 42.81% exhibited cytotoxic effects, while 57.19% did not. Drug-Induced Liver Injury (DILI) toxicity results indicated a low hepatotoxicity rate, with only 11.07% of compounds testing positive and 88.93% negative. AMES toxicity testing for mutagenic potential revealed that 18.08% were mutagenic, while 81.92% were not. The hepatic microsomal (HLM) stability assay showed that 91.63% of the compounds were metabolically stable, with only 8.37% exhibiting instability. Furthermore, blood–brain barrier (BBB) permeability analysis indicated that 41.19% of the compounds could penetrate the BBB, suggesting possible central nervous system activity, while 58.81% were non-permeable as shown in Figure 2g. These findings underscore the potential suitability of natural compounds for further drug development, highlighting their promising pharmacokinetic profiles and suggesting avenues for continued exploration in pharmaceutical research.

#### 2.2.2. Predicting Drug Likeness with Data Warrior for the IMPPAT Compounds

Drug-likeness prediction was conducted for compounds selected from the IMPPAT database as potential SVMP inhibitors. Approximately 64.23% of the natural compounds exhibited favorable drug-likeness scores based on their physicochemical properties. Notably, the molecular weight descriptor emerged as a significant discriminator, with natural compounds ranging from 56.06 to 2504.24 g/mol and a mean of 348.5 g/mol. Small natural compounds with high drug-likeness scores may hold promise for fragment-based drug discovery in future investigations.

### 2.3. Preliminary Toxicity Assessment of Natural Compounds: Evaluating Safety Parameters

Early toxicity prediction is essential for identifying potential safety issues with successful compounds. In this study, DataWarrior and ADMEpred predictions were used to evaluate the activity profiles of natural substances. The toxicity profiles of most natural compounds were encouraging; as shown in Figure 3a, most natural compounds fell within safe limits for AMES toxicity (81.92%), bore minimal risk of mutagenicity (85.82%), and were unlikely to be tumor-promoting agents (89.42%) or irritants (70.8%). A comprehensive pharmacokinetics analysis of natural compounds was conducted, with detailed cut-off points illustrated. Ninety-one compounds with favorable drug scores were subsequently chosen for further analysis, as illustrated in Figure 3b.

### 2.4. Binding Site Prediction

The Atrolysin protein is depicted as a cartoon, with red circles indicating putative interacting residues in surface representation. An enlarged view highlights residues within a 5.0 Å distance, displayed in cyan, suggesting possible interacting residues. Notable residues within this vicinity include LEU-106, THR-107, GLY-108, GLY-109, LEU-110, ALA-111, VAL-138, THR-139, HIS-142, GLY-143, HIS-146, GLU-151, HIS-152, ILE-165, ARG-167, PRO-168, GLY-169, and LEU-170. These residues are likely to play significant roles in ligand binding and protein–ligand interactions shown in Figure 4, thereby influencing the functional properties of the Atrolysin protein. The binding site was determined using the position of the co-crystallized ligand as a reference, ensuring accuracy in the identification of key residues involved in ligand interaction.

### 2.5. Molecular Docking Analysis

Virtual screening techniques are widely utilized in drug discovery. In this study, we conducted a virtual screening of 91 small molecules from the IMPATT database. To validate our results, we analyzed data for these compounds using three tools—AD4, ADZn, and AutoDock—which are presented in Appendix A. This analysis revealed varying binding affinities among seven selected compounds toward the protein structure 1ALT. The top ten ligands with the highest binding energies and their respective amino acids involved in the interactions are listed in Table 1, while the chemical structures of these compounds are depicted in Appendix A. The molecular docking results offer insights into the binding interactions for each ligand, as shown in Figure 5. Ligand IMPHY003833 formed hydrogen bonds with residue 142 (HIS) and exhibited hydrophobic interactions with residues 108 (LEU) and 138 (VAL), yielding a binding free energy of −7.76 kcal/mol and an estimated inhibition constant (Ki) of 2.04 µM. Ligand IMPHY011953 displayed hydrogen bonding with residue 142 (HIS) and significant hydrophobic interactions with residues 108 (LEU) and 170 (LEU), resulting in a binding free energy of −8.97 kcal/mol and a Ki of 266.13 nM. Ligand IMPHY003823 engaged in hydrogen bonding with residue 152 (HIS) and showed hydrophobic interactions with residues 108 (LEU) and 170 (LEU), leading to a binding free energy of −8.19 kcal/mol and a Ki of 999.23 nM. Ligand CID5291 exhibited extensive hydrophobic interactions with residues 108 (LEU) and 170 (LEU), contributing to a binding free energy of −7.81 kcal/mol and a Ki of 1.89 µM. Ligand IMPHY001495 interacted with residues 107 (THR) and 170 (LEU), resulting in a binding free energy of −8.66 kcal/mol and a Ki of 449.31 nM. Ligand IMPHY008983 demonstrated interactions with residues 142 (HIS) and 170 (LEU), yielding a binding free energy of −8.51 kcal/mol and a Ki of 574.27 nM. Ligand CID 5362422 exhibited various interactions, including hydrogen bonding with residue 142 (HIS) and hydrophobic interactions with residues 108 (LEU) and 143 (GLU), leading to a binding free energy of −8.0 kcal/mol and a Ki of 1.36 µM. Ligand IMPHY014737 interacted with residues 107 (THR) and 170 (LEU), resulting in a binding free energy of −8.54 kcal/mol and a Ki of 548.38 nM. Ligand IMPHY008176 showed interactions with residues 108 (LEU) and 138 (VAL), yielding a binding free energy of −8.33 kcal/mol and a Ki of 788.08 nM. Finally, ligand CID132519 demonstrated interactions with residues 142 (HIS) and 170 (LEU), contributing to a binding free energy of −8.39 kcal/mol and a Ki of 703.53 nM.

### 2.6. DFT Analysis

The density functional theory (DFT) results provide the electronic properties, potential reactivity, and stability of various compounds. Each compound’s highest occupied molecular orbital (HOMO) and lowest unoccupied molecular orbital (LUMO) energies, along with their band gap, reveal their electronic structure and potential for electron transfer processes shown in Figure 6 and the electronic properties shown in Table 2. The compounds IMPHY011953 and IMPHY014737 exhibit relatively small band gaps of 0.1792 and 0.132 eV, respectively, indicating their potential as conductive materials. Furthermore, the chemical potential and hardness values provide information about the compounds’ ability to donate or accept electrons and their resistance to changes in electron density. Compounds CID5291 and IMPHY00383 have relatively high chemical hardness, suggesting stability and resistance to electronic perturbations. Electronegativity values offer insights into the compounds’ ability to attract electrons in a chemical bond; compounds like pareirine and CID5291 exhibit moderate electronegativity, indicating balanced electron attraction tendencies. The electrophilicity index indicates the compounds’ susceptibility to nucleophilic attack, with higher values indicating increased susceptibility. Compounds CID132519 and IMPHY008176 exhibit relatively high electrophilicity indices, suggesting their potential reactivity toward nucleophiles. Moreover, the dipole moment reflects the compounds’ overall polarity, with higher values indicating greater polarity.

### 2.7. Molecular Dynamics Analysis of Protein–Ligand Complexes

Molecular dynamics (MD) analysis is a well-established computational simulation technique that allows for the study of the physical movements of atoms and molecules, providing dynamic data at atomic spatial resolution. In this study, we optimized the structures of both the Apo 1ALT protein and its complex with flavonoids, ensuring they achieved minimum potential energy and maximum force values. We performed 100-nanosecond simulations for both the Apo protein and the flavonoids-1ALT complex to assess their stability. The MD simulation results were analyzed using several parameters, including hydrogen bonds (H-bonds), radius of gyration, root mean square deviation (RMSD), and root mean square fluctuation (RMSF). These parameters were computed throughout the entire simulation trajectory, allowing for a comparative evaluation of the stability and dynamic properties of both the Apo protein and the protein–ligand complex, as detailed in Figure 7.

#### 2.7.1. Root Mean Square Deviation (RMSD)

The RMSD values for the analyzed compounds provide significant insights into the structural stability of protein–ligand complexes over the simulation period. The maximum RMSD values range from 0.1628004 to 0.2763315, indicating varying degrees of deviation from the initial protein structure. Notably, IMPHY003833 exhibits the lowest maximum RMSD (0.1628004), suggesting superior structural stability of protein–ligand complex compared to other compounds’ protein–ligand complex. The minimum RMSD values are consistently low across all compound complexes, with CID5291 showing the least deviation at 0.0004933, indicating periods of very stable structural alignment. The running average and probability distribution are shown in Appendix A, and ligand RMSD is shown in Appendix A.

#### 2.7.2. Root Mean Square Fluctuations (RMSF)

RMSF analysis highlights the flexibility of different regions within the protein structures. Maximum RMSF values range from 0.1849 to 0.6517, with IMPHY003833 demonstrating the highest fluctuation. In contrast, CID5291 shows the lowest maximum RMSF (0.1849), indicating a more stable and less flexible protein structure. The minimum RMSF values, ranging from 0.0339 to 0.0412, suggest regions with minimal atomic movement, with CID5291 again exhibiting the least fluctuation.

#### 2.7.3. Radius of Gyration (Rg)

The radius of gyration (Rg) values provide insights into the overall compactness of the protein structures. The maximum Rg values range from 1.61477 to 1.63432, indicating slight variations in protein compactness. IMPHY014737 shows the highest maximum Rg (1.63432), suggesting the most compact state, while IMPHY008176 has the lowest maximum Rg (1.61477). The minimum Rg values range from 1.56411 to 1.57648, with IMPHY008983 demonstrating the most expanded state and IMPHY003833 the most compact. The running average and probability distribution are shown in Appendix A.

#### 2.7.4. Hydrogen Bond Analysis

The hydrogen bond analysis reveals dynamic interactions within the protein structures. The maximum number of hydrogen bonds ranges from 2 to 6, with CID5362422 forming the highest number of bonds (6), suggesting strong and stable interactions. All compounds exhibit a minimum of 0 hydrogen bonds, indicating fluctuating interaction patterns during the simulation. Although weaker than ionic and covalent bonds, intermolecular hydrogen bonds (H-bonds) significantly contribute to the complex formation and stability of the ligand–protein structure. Throughout the simulation period depicted in Figure 6c, the complex maintained a consistent range of intermolecular H-bond interactions with the receptor protein, ranging between 0 and 3. The average number of H-bond interactions observed was approximately 2.

### 2.8. Binding Energy Analysis

The binding energy analysis of the compounds reveals significant variations in their interaction energies, highlighting their potential effectiveness in molecular dynamics simulations. IMPHY011953 exhibits moderate binding energy with notable contributions from both van der Waals and electrostatic interactions, making it a compound of interest due to its balanced interaction profile. IMPHY003823 has relatively low binding energy, characterized by weaker interactions. CID5291 shows significant binding energy primarily driven by strong van der Waals interactions, despite a positive electrostatic contribution. Similarly, IMPHY001495 demonstrates substantial binding energy with notable van der Waals interactions and positive electrostatic energy, indicating a robust binding profile. IMPHY008983 displays moderate binding energy with contributions from both interaction types, suggesting it as a potential phytochemical compound. CID132519 demonstrates considerable binding energy driven by strong van der Waals interactions and positive electrostatic contributions. IMPHY014737 exhibits notable binding energy characterized by strong van der Waals and significant electrostatic interactions, highlighting its potential effectiveness. IMPHY008176 presents moderate binding energy contributions from both interaction types, making it another compound of interest. CID5362422 shows notable binding energy with significant van der Waals and strong electrostatic interactions, suggesting a stable and effective binding profile. IMPHY003833 exhibits binding energy driven by strong van der Waals interactions and moderate electrostatic interactions, indicating it as a promising candidate. These results suggest that compounds CID5291, IMPHY001495, IMPHY008176, IMPHY014737, IMPHY008983, and IMPHY003833 are top candidates due to their favorable binding energies and strong interaction forces, as shown in Figure 8a and Table 3.

The binding affinities of these compounds to specific amino acid residues of the protein reveal significant interactions at key residues. CID5291 exhibits the strongest binding affinity with GLU-106, showing a value of −10.4581, indicating a particularly strong interaction. In contrast, CID132519 shows a much weaker interaction with GLU-106, having a value of −1.0499. For LEU-108, IMPHY003833 demonstrates a strong binding affinity of −6.6726, whereas CID132519 again shows a weaker interaction with LEU-108 at −0.5642. Regarding the residue GLY-137, binding affinities are minimal, with values close to zero, such as −0.008 for CID5291 and −0.0076 for IMPHY001495, suggesting that GLY-137 may not play a critical role in binding. HIS-142 shows diverse binding affinities among the compounds, with IMPHY003823 exhibiting a positive affinity of 2.8526, indicating a weak interaction, while IMPHY008176 shows strong binding with HIS-142 at −3.1525. Finally, for PRO-168, IMPHY003823 shows a very strong binding affinity of −5.1471, whereas ID132519 and CID5362422 exhibit weaker interactions with values of −1.7573 and −2.0837, respectively. These results are shown in Figure 8b.

## 3. Discussion

Developing new drug candidates is a complex and costly long-term process. However, historical evidence underscores the significant contribution of natural products research in addressing public health challenges, particularly neglected conditions such as snakebite envenomation. Plants have long served as crucial sources of medicinal compounds capable of inhibiting venom toxins [13].

SVMPs play a critical role in the envenomation process, resulting in local and systemic effects such as hemorrhage, necrosis, blisters, and inflammation. Their proteolytic activity involves cleaving and degrading essential basement membrane components such as laminin, nidogen, fibronectin, proteoglycans, and type IV collagen. This degradation disrupts connective tissues, compromising the structural integrity of blood vessels and leading to hemorrhage. SVMPs also contribute to coagulation disorders by cleaving coagulation factors, thereby inducing a procoagulant state and affecting platelets, resulting in consumption coagulopathy and subsequent bleeding [26,27,28,29,30,31]. Furthermore, SVMPs directly stimulate leukocytes and influence inflammatory components of the complement system, promoting the release of damage-associated molecular patterns (DAMPs) through their proteolytic actions [32,33,34,35].

Plant-derived therapeutics are increasingly recognized for their potential in treating snakebite envenoming [36,37,38]. In developed countries, plant-based drugs constitute approximately 25% of the total therapeutic drugs, whereas in countries such as China and India, this percentage rises to 80% [39,40]. Phytochemicals derived from plants, whether in their natural form or as derivatives, have been employed as adjunctive treatments alongside antivenom. These plant extracts contain a variety of bioactive compounds capable of mitigating venom toxicity, albeit with limited efficacy. Purified phytochemicals, however, may offer more potent therapeutic effects [41,42,43,44,45].

The present study highlights the potential of plant-derived compounds as candidates for snakebite envenoming treatment, with a particular focus on their molecular interaction with venom proteins, such as Atrolysin Metalloproteinase (AMP). Our findings, incorporating molecular docking, molecular dynamics (MD) simulations, and density functional theory (DFT) calculations, suggest that these compounds could be effective in neutralizing venom toxicity. Notably, several phytochemicals, including Corytuberine (IMPHY001495), Akuammigine (IMPHY011953), and Imatinib (CID5291), demonstrated strong binding energies and favorable interaction profiles with venom proteins. These results align with previous studies showing that plant-based compounds can inhibit venom activity through direct protein interaction [36,38,39,46,47,48]. Our molecular docking results revealed significant binding affinities for the compounds, with the best-performing compounds—such as IMPHY011953, IMPHY001495, and CID5291—exhibiting docking scores ranging from −8.66 to −8.97 kcal/mol. These binding affinities are particularly noteworthy when compared to previous studies that identified plant-derived compounds with binding energies between −7.0 and −8.0 kcal/mol [49,50,51].

The electronic properties of the compounds through DFT calculations provided additional insights into their potential for therapeutic application. The HOMO-LUMO gaps of the compounds varied, with most exhibiting relatively small gaps indicative of good electrophilic properties. IMPHY001495 had a HOMO of −0.20577 and a LUMO of −0.0195, resulting in a gap of 0.1863 eV, which indicates a high potential for electron donation and stable binding interactions (Table 1). Compounds with smaller HOMO-LUMO gaps tend to exhibit better stability and higher interaction energies in biological systems. The chemical potential and electronegativity of these compounds further suggest that they are likely to interact strongly with the charged residues of the target venom proteins, as supported by the high dipole moments observed in compounds like IMPHY003833 and CID5291 (Table 1) [52,53,54,55].

The molecular dynamics simulations indicated that these compounds form stable complexes with the target proteins, as evidenced by the steady RMSD values observed during the simulations. Compounds like IMPHY011953 and CID5291 showed minimal RMSD fluctuations, signifying strong complex stability. This finding complements the research by [56] those who utilized MD simulations to evaluate the stability of venom-neutralizing agents. In particular, the consistent stabilization of RMSD values in our study further supports the notion that these compounds could have prolonged interactions with venom proteins, thus enhancing their potential therapeutic application.

Plant-derived compounds with binding energies in a similar range but without a detailed electronic analysis go further by incorporating both MD simulations and DFT to predict the stability and reactivity of these compounds. IMPHY001495, IMPHY011953, and CID5291 demonstrated robust binding energy MMPBSA calculations, which is significantly stronger than the results reported for similar compounds [57]. The MMPBSA results from our study confirm the strength of the interactions, showing strong van der Waals interactions and moderate electrostatic contributions. This suggests that the compound might offer a more stable and effective solution for venom inhibition compared to some previously studied compounds [58,59].

The analysis of 4708 unique compounds from the IMPPAT database identified 91 phytochemicals exhibiting drug-like properties based on molecular descriptors and ADME profiling. Subsequent molecular docking studies revealed that compounds IMPHY001495-phytochemical name: Corytuberine, IMPHY014737-phytochemical name: 16-Hydroxy-16,22-dihydroapparicine, IMPHY011953-phytochemical name: Akuammigine, IMPHY008983-phytochemical name: Pareirine, IMPHY008176-phytochemical name: Benzenamine, *N*, *N*-dimethyl-2-(1,2,3,9-tetrahydropyrrolo [2,1-b]quinazolin-3-yl), and CID5291 pubchem name Imatinib exhibited strong binding affinities to Atrolysin Metalloproteinase, with binding energies ranging from −8.7 to −10.2 kcal/mol. Detailed interaction analyses indicated that these compounds formed stable hydrogen bonds and hydrophobic interactions with key residues within the active site of the protein. Molecular dynamics (MD) simulations further demonstrated the dynamic stability of the protein-ligand complexes, with RMSD values stabilizing after an initial fluctuation period. Density functional theory (DFT) calculations provided additional insights into the electronic properties and stability of the identified compounds.

This study investigated compounds derived from medicinal plants known for their effectiveness in treating snake venom, specifically Akuammigine from Alstonia scholaris, *Mitragyna parvifolia*, and *Rauvolfia serpentina*; Alstonine from *Rauvolfia serpentina*; Benzenamine, *N*,*N*-dimethyl-2-(1,2,3,9-tetrahydropyrrolo [2,1-b]quinazolin-3-yl)- from *Justicia adhatoda*; Corytuberine and Pareirine from *Cissampelos pareira*; Serpentine from *Rauvolfia serpentina*; and 16-Hydroxy-16,22-dihydroapparicine from *Tabernaemontana divaricata* [60,61,62,63,64,65].

## 4. Materials and Methods

### 4.1. Identification of Medicinal Plants and Bioactive Molecules

In this study, we utilized the Indian Medicinal Plants, Phytochemistry And Therapeutics (IMPPAT) database as the primary source for identifying phytochemical compounds with the potential to address snake bite envenomation. From IMPPAT 2.0, we manually extracted information to identify relevant plants for snakebites and their associated phytochemicals. This database provides comprehensive data on 17,967 phytochemicals linked to 408 Indian medicinal plants, establishing its prominence as the most extensive digital repository [66,67]. Additionally, to enhance our analysis, we included nine well-known metalloproteinase inhibitors from the PubChem database.

### 4.2. Comprehensive Evaluation and Prediction of Drug-like Compound Properties

The concept of drug-like chemical compounds is essential to the fields of drug discovery and candidate selection. It refers to substances that have pharmacokinetic properties, making them suitable for progression into human-phase clinical trials. We comprehensively assessed various physicochemical characteristics, pharmacokinetic profiles, drug similarity, and medicinal chemistry features of compounds, including traits such as lipophilicity and water solubility, utilizing ADMETlab (https://admetmesh.scbdd.com/ accessed on 23 November 2023). ADMETlab predicted significant factors such as physicochemical characteristics, lipophilicity, pharmacokinetics, and compliance with Veber’s filter and Lipinski’s rule of five requirements [68]. The variable nearest neighbor (vNN) approach was used to anticipate factors linked to metabolism, such as the human liver microsomal test [69]. The phytochemicals’ water solubility, LogP, and LogS values were obtained from IMPPAT 2.0 and forecasted using ALOGPS 2.1 and RDkit (https://www.rdkit.org/ accessed on 2 December 2023) Additionally, DataWarrior tools v6.0 were used to compute compound properties (CPs), including cLogP (calculated logarithm of the partition coefficient), MW (molecular weight), TPSA (topological polar surface area), and RB (rotatable bonds), and the tally of hydrogen bond donors and acceptors (HBD/HBA). These tools predict mutagenicity, tumorigenicity, irritancy, drug-likeness, and drug scores, with visualization features augmenting the comprehension of these properties, thereby facilitating the identification of the most promising candidates. In terms of exclusion criteria, compounds were evaluated using specific thresholds for each parameter, ensuring that only those with suitable drug-like properties were retained. Excluded compounds were those that did not meet the minimum requirements. The Drug-Likeness Score is calculated as the sum of individual fragment scores based on the presence of substructure fragments within a compound, while the Drug Score (ds) integrates various physicochemical properties, drug-likeness, and toxicity risks using the formula ds = Π(s_i_^x^_i_). Here, s_i_ represents the contributions of properties like cLogP and molecular weight, and t_i_ indicates toxicity risk levels [66,70].

### 4.3. Ligand Preparation

The ligand library comprising potential inhibitor molecules against Atrolysin Metalloproteinase was sourced from the DataWarrior tool v6.0. The three-dimensional (3D) structures of phytochemicals were initially provided in SDF format and subsequently converted to PDBQT format using Open Babel v3 [71]. Phytochemicals lacking 3D structural data were excluded from the study. In total, 91 compounds with favorable drug scores were retrieved from the IMPPAT database.

### 4.4. Protein Structure Preparation

The protein structure has a resolution of 1.80 Å and consists of 207 amino acids forming homodimers. The downloaded crystal structure 1ATL contains 202 amino acids (residue numbers 192 to 393) and includes one Zn ion. Notably, certain residues at the *N*-terminal and C-terminal were unresolved; however, their absence does not impact the binding interactions as they are distant from the active site. The protein structure was prepared by adding hydrogen atoms and subsequently subjected to energy minimization using UCSF-Chimera software v1.18 [72,73]. This software was also employed for refining the structure and minimizing the energy of the selected protein.

### 4.5. Molecular Docking

Molecular docking techniques are pivotal in drug discovery, aiding in pose prediction, virtual screening, and estimating ligand-protein binding affinity. Due to the known limitations of docking scoring functions in providing accurate binding energies, especially for chemically diverse compounds, we have utilized AutoDock Vina primarily as a tool for identifying potential binding poses rather than for precise affinity comparison. The protein structure was prepared by converting from PDB to PDBQT format using Open Babel software v3, and Kollman charges were applied to add polar hydrogens and charges. Grid points were assigned around the protein with XYZ coordinates set at 40 Å × 25 Å × 40 Å, centered at 3.000 × 25.000 × 5.068, and a spacing of 0.375 Å. During the docking process, ten poses were generated for each ligand, and these poses underwent clustering to determine the optimal binding pose based on binding energy and root mean square deviation (RMSD) criteria, elucidating atomic interactions. To validate the docking results, a Python script (https://vina.scripps.edu/manual/ (accessed on 10 January 2024)) was used to identify compounds with high docking scores, indicating minimal binding energy [74,75]. Interaction analysis between protein-ligand complexes was conducted using PLIP [76], with known metalloproteinase inhibitors docked as controls.

### 4.6. Molecular Properties Through Density Functional Theory (DFT) Analysis

Density functional theory (DFT) serves as a computational quantum mechanical tool essential for correlating calculated molecular energies and assessing chemical activity. Utilizing the B3LYP method and Gaussian 09 with a 6-31G (d, p) basis set, we computed various molecular properties, including the lowest unoccupied molecular orbital (LUMO), highest occupied molecular orbital (HOMO), energy gap, total energy, and chemical potential. Pivotal global descriptors crucial for understanding molecular behavior, such as softness, absolute hardness, electrophilicity index, and electronegativity, were meticulously determined. By analyzing the electron density in molecular orbitals, our study provided invaluable insights into the structure-activity relationship of the molecules. This investigation is deeply rooted in the theoretical framework of density functional theory (DFT), founded on the Hohenberg–Kohn theorem, which facilitates a profound understanding of molecular properties. Our computational DFT study generated a comprehensive range of approximately ten distinct molecular descriptors to delineate molecular activity, including parameters such as total energy, dipole moment, and HOMO/LUMO values extracted using Gaussian 09W program. Additional descriptors such as HOMO/LUMO gap, absolute hardness, global softness, electronegativity, chemical potential, and electrophilicity index were computed using specific mathematical formulas, contributing to a thorough characterization of molecular behavior [77,78,79].

### 4.7. Molecular Dynamics Simulation

Molecular dynamics (MD) simulation is a computational tool used to predict molecular movements and behaviors. The top ten compounds with the highest binding energies and best poses were selected for MD simulations using the Groningen Machine for Chemical Simulations (GROMACS). In our research, MD simulations focused on exploring the interactions between these compounds and Atrolysin C. The best docking poses were chosen to investigate these interactions in detail [80]. The protein and ligands were prepared using the CHARMM36 force field and Avogadro program, respectively. The CHARMM36 force field was specifically chosen for protein preparation, while the CGenFF server and Avogadro program were utilized for ligand preparation. Subsequently, complexes and their topologies were generated. Solvation was performed for all complexes, and ions were added as necessary to neutralize the systems [81,82]. Molecular electrostatic interactions were accurately estimated using ME algorithms, with parameter settings including a PME order of 4, Fourier spacing of 0.16, and a temperature scale set at 310 K. Further equilibration steps included thermal equilibration (NVT step) for 1 ns, followed by NPT simulation at 300 K using a modified Berendsen thermostat for temperature coupling (T-coupling) and V-rescale for pressure coupling (P-coupling). MD simulations were conducted over 100 ns without any position restraints, tracking system dynamics with outputs recorded every 2 ps and velocities and coordinates stored every 10 ps (nst-vout and nstxout). Precise computational parameters, including Coulombic and van der Waals force cut-offs set at 1.0 nm (rcoulomb and rvdw), along with the utilization of Particle Mesh Ewald (PME) and periodic boundary conditions (PBC), ensured accurate calculation of long-range electrostatic interactions. Finally, MD simulations were performed on three selected protein-ligand complexes for 100 nanoseconds. Throughout the MD trajectories, analyses such as root mean square deviation (RMSD), root mean square fluctuation (RMSF), hydrogen bonds, and radius of gyration were conducted for the top ten compounds within the protein-ligand complexes [83].

### 4.8. Analysis of Binding-Free Energy

The molecular mechanics Poisson–Boltzmann surface area (MM-PBSA) method is extensively employed for calculating interaction energies in biomolecular complexes, integrating conformational fluctuations and entropic effects, as discussed [84]. In our investigation, we focused on the final 20 nanoseconds of simulation trajectories for selected compounds using the g_mmpbsa module within GROMACS. This module computes the binding free energy (BFE) between proteins and ligands. The MM-PBSA approach decomposes the free energy into components such as van der Waals and electrostatic interactions (ΔEvdW and ΔEele), as well as polar and nonpolar solvation energies (ΔGpol and ΔGnonpol), incorporating temperature and entropic contributions (TΔS) [84].

## 5. Conclusions

The pharmacoinformatic studies, employing molecular docking, DFT analysis, molecular dynamics simulations, and binding energy calculations, effectively investigated the interactions between phytochemical compounds from medicinal plants and Atrolysin. These analyses provided crucial structural insights into the toxin–ligand interactions, identifying several compounds with potential as antidotes against Atrolysin. This study represents a comprehensive pharmacoinformatic approach identifying plant-derived inhibitors for the metalloproteinase Atrolysin from Crotalus atrox venom. The identified compounds exhibit promising drug-like properties, favorable ADME (Absorption, Distribution, Metabolism, and Excretion) profiles, and strong binding affinities, positioning them as promising candidates for further development as alternative treatments for snakebite envenomation. The findings contribute valuable insights into protein-ligand interactions and lay the groundwork for future in vivo and clinical investigations.

## Figures and Tables

**Figure 1 ijms-25-12675-f001:**
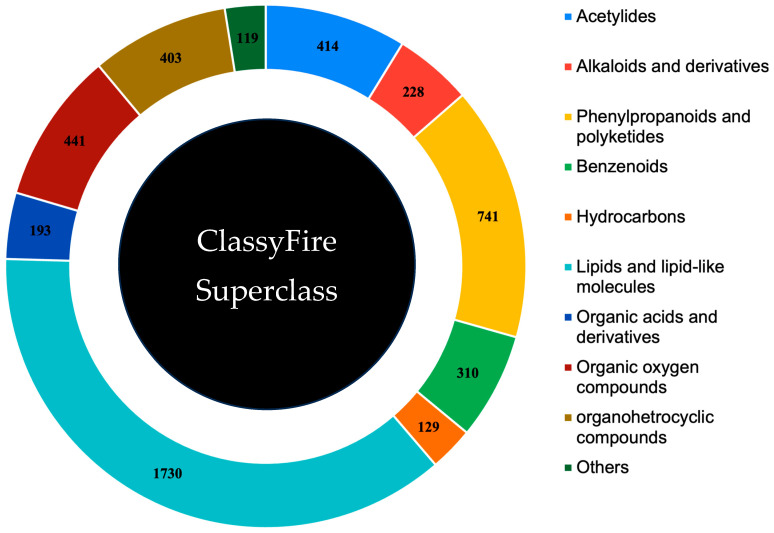
ClassyFire classification: A diverse array of phytochemicals manually curated from the IMPPAT database.

**Figure 2 ijms-25-12675-f002:**
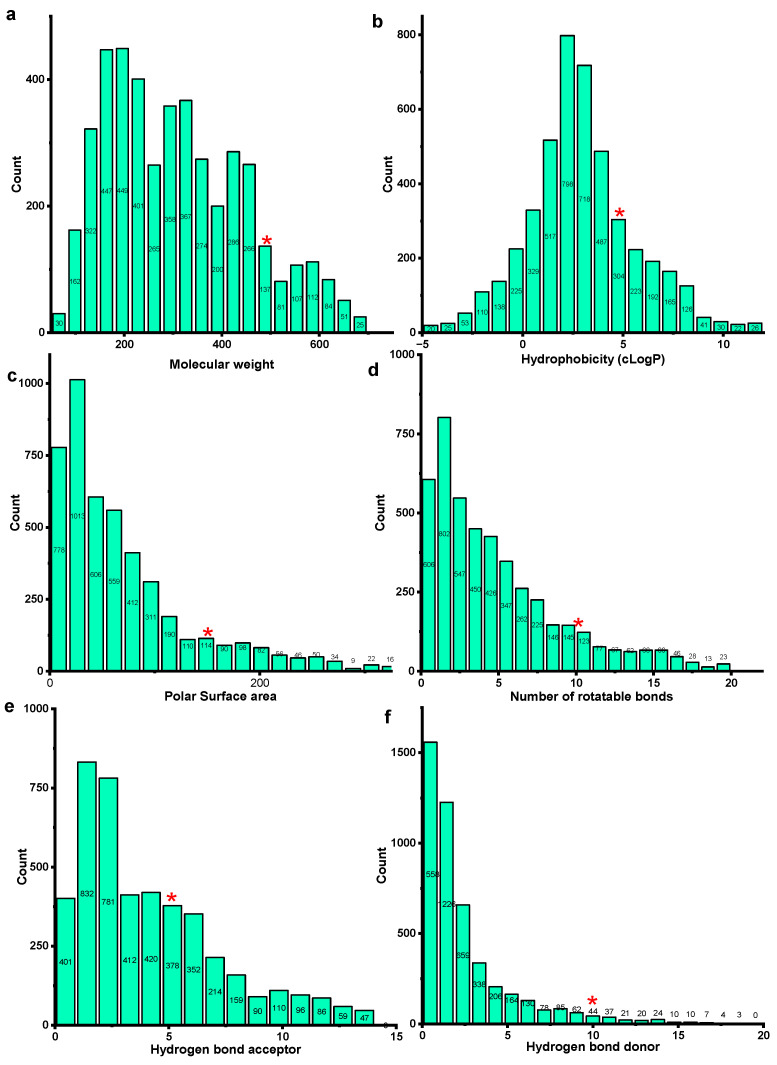
Exploring the chemical space of compounds as inhibitors of SVMP. Analysis of the physicochemical properties of compounds acting as inhibitors of SVMP, including (**a**) molecular weight, (**b**) hydrophobicity (cLogP), (**c**) topological polar surface area, (**d**) hydrogen-bond acceptors, (**e**) hydrogen-bond donors, and (**f**) number of rotatable bonds. In (**a**–**f**), the red asterisk signifies the cut-off points established by Lipinski and Veber rules. (**g**) Toxicity assessments conducted using the vNN tool encompass cytotoxicity, DILI toxicity, AMES toxicity, HLM stability assay, and BBB permeability.

**Figure 3 ijms-25-12675-f003:**
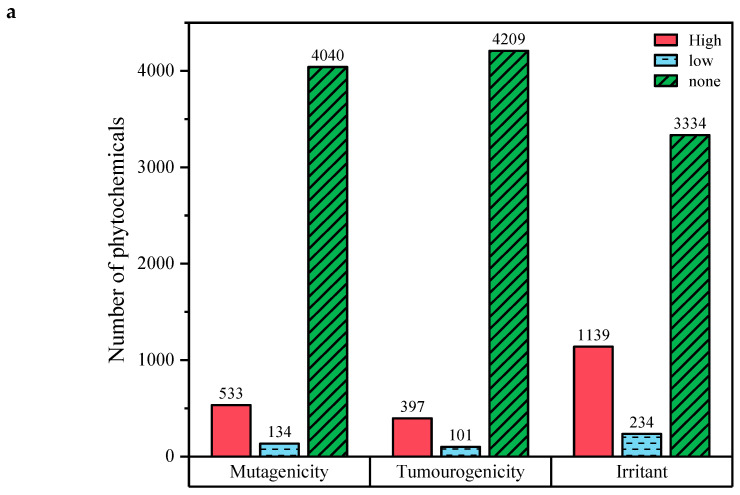
Depicts the DataWarrior tool v6.0, showcasing (**a**) predictions for mutagenicity, tumorigenicity, and irritant properties and (**b**) selected compounds based on their drug scores for further analysis.

**Figure 4 ijms-25-12675-f004:**
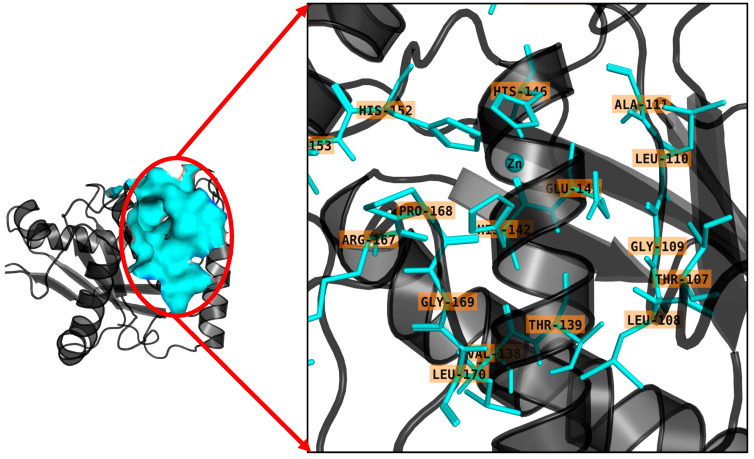
The Atrolysin protein is depicted as a cartoon with red circles indicating putative interacting residues in surface representation. An enlarged view highlights residues within a 5.0 Å distance, displayed in cyan, suggesting possible interacting residues.

**Figure 5 ijms-25-12675-f005:**
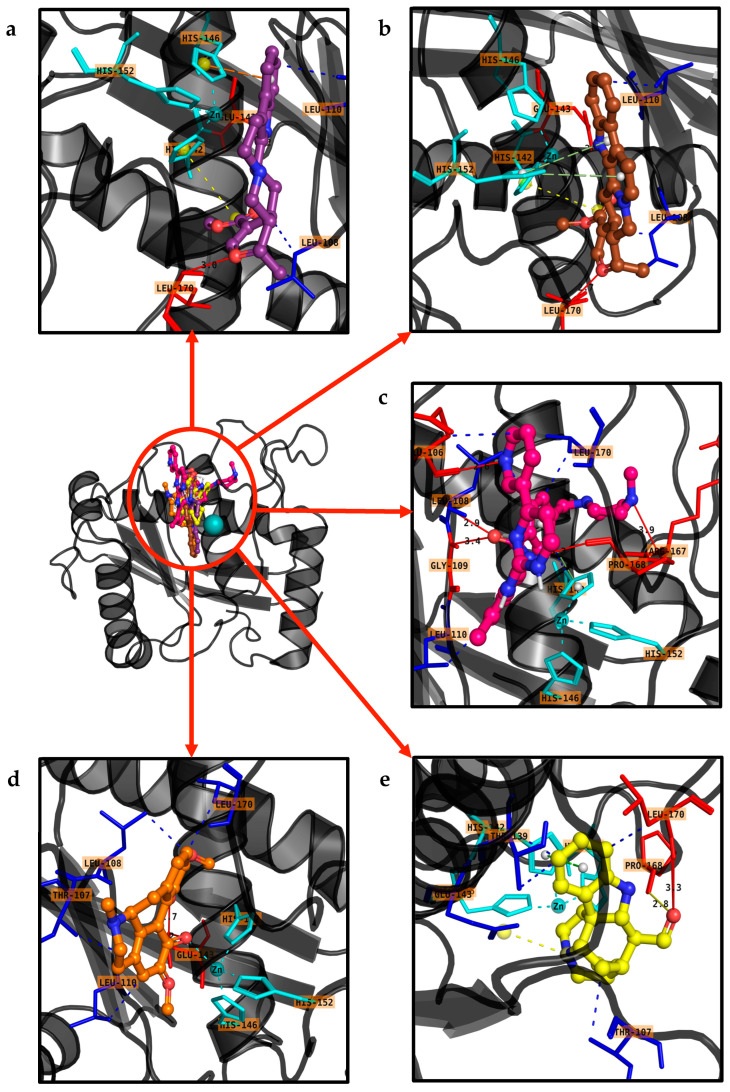
Molecular docking studies of natural compounds against the Atrolysin protein were performed and validated using AutoDock Vina v1.2 and PyMOL v2.5.7. The top 10 compounds selected for analysis were visually represented in PyMOL, each assigned a distinct color for clarity: (**a**) IMPHY011953-Purple, (**b**) IMPHY003823-Brown, (**c**) CID5291-Hotpink, (**d**) IMPHY001495-TV Orange, (**e**) IMPHY008983-TV Yellow, (**f**) CID5362422-Limegreen, (**g**) IMPHY014737-Grey80, (**h**) IMPHY008176-Marine, (**i**) CID132519-Cyan, (**j**) IMPHY003833-Sky-blue.

**Figure 6 ijms-25-12675-f006:**
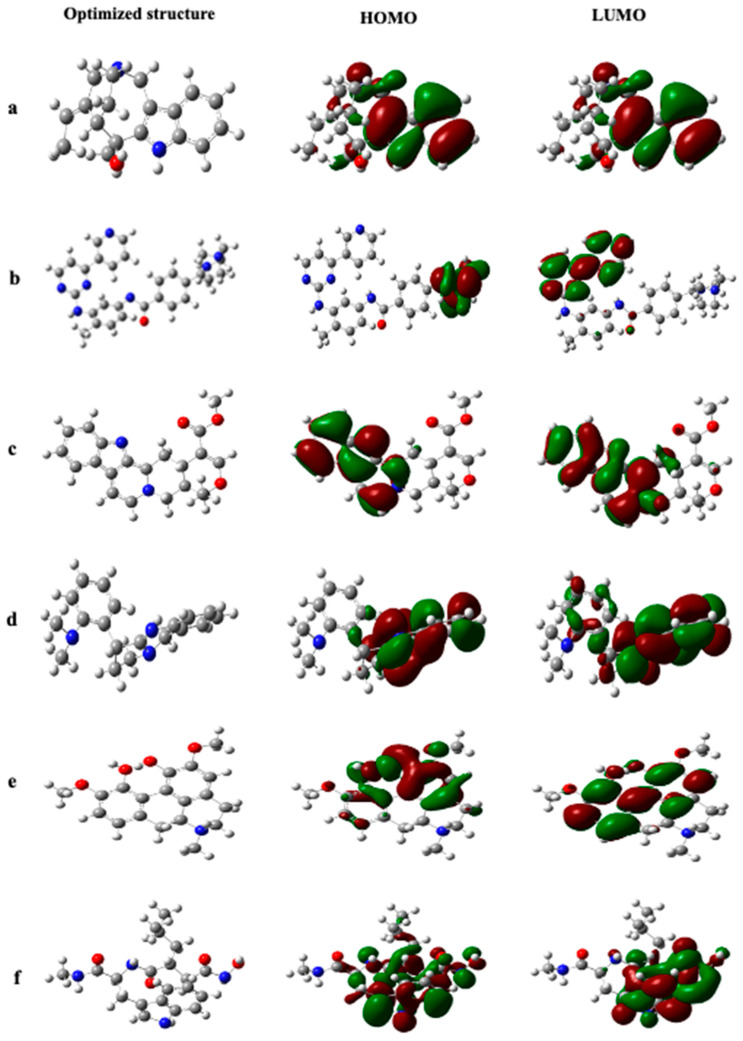
Electron density maps of the Highest Occupied Molecular Orbital (HOMO) and Lowest Unoccupied Molecular Orbital (LUMO) were generated for the top phytochemicals identified as follows: (**a**) IMPHY011953, (**b**) IMPHY003823, (**c**) CID5291, (**d**) IMPHY001495, (**e**) IMPHY008983, (**f**) CID5362422, (**g**) IMPHY014737, (**h**) IMPHY008176, (**i**) CID132519, (**j**) IMPHY003833.

**Figure 7 ijms-25-12675-f007:**
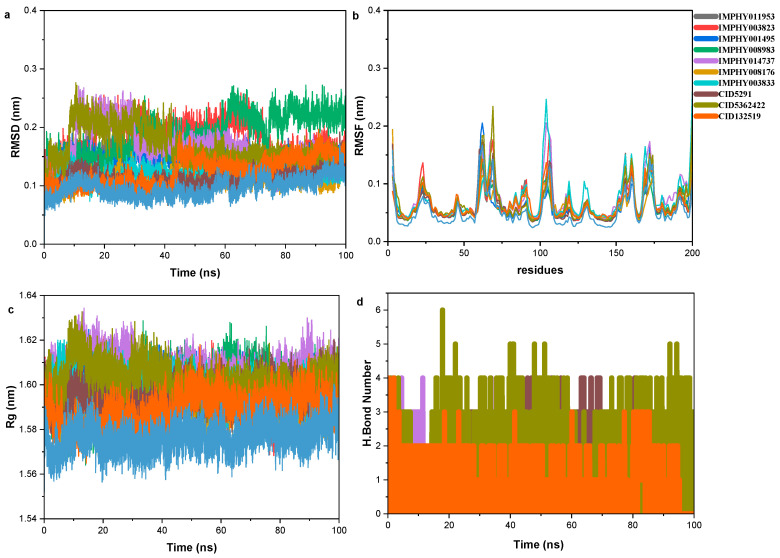
Results from molecular dynamics (MD) simulations of the ten selected compounds, analyzing their interactions with the binding site over a 100 ns simulation trajectory: (**a**) Root mean square deviation (RMSD) for the complexes, (**b**) root mean square fluctuation (RMSF), (**c**) radius of gyration, and (**d**) hydrogen bond analysis. These analyses provide insights into the stability, flexibility, compactness, and specific interactions of the compounds with the binding site.

**Figure 8 ijms-25-12675-f008:**
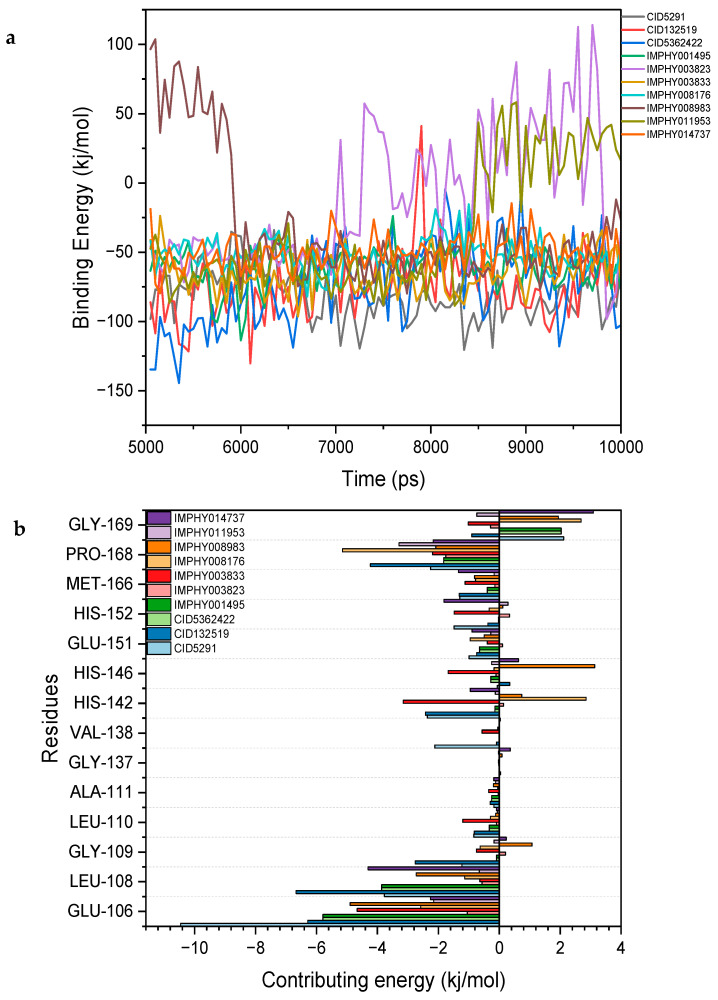
The per-residue energy decomposition and binding energy analysis of compounds: (**a**) Contribution of amino acid residues with the highest binding energy to the interaction with the compounds; (**b**) top contributing amino acid residues to the binding energy in complexes with the ten most potent compounds.

**Table 1 ijms-25-12675-t001:** Binding affinity scores, interacting residues, and distances for hydrogen and hydrophobic interactions in molecular docking studies of natural compounds against Atrolysin protein.

Compound ID	Metal Complexes	Hydrophobic Interactions	Hydrogen Bonds	Distance H-A (Å)	Distance D-A (Å)	π–Cation Interactions	Ligand Group	Dock Score Kcal/mol	Ligand Group
IMPHY011953	146 HIS, 146 HIS, 152 HIS	108 LEU, 110 LEU	143 GLU	2.27	3.22	146 HIS	Aromatic	−8.97	Carboxylate
			170 LEU	2.06	3				
IMPHY003823	146 HIS, 146 HIS, 152 HIS	108 LEU, 110 LEU	143 GLU	2.38	3.36	152 HIS	-	−8.19	Carboxylate
			170 LEU	1.89	2.67				
			106 GLU	2.06	2.64				
CID5291	146 HIS, 146 HIS, 152 HIS	106GLU, 108 LEU, 110 LEU, 142 HIS, 170 LEU	108 LEU	1.95	2.9	142 HIS	-	−7.81	-
			109 GLY	2.55	3.38				
			167 ARG	3.3	3.93				
			168 PRO	2.22	3.25				
IMPHY001495	UNL, 146 HIS, 146 HIS, 152 HIS	107 THR, 108 LEU, 110 LEU, 170 LEU	143 GLU	1.9	2.66		-	−8.66	-
			143 GLU	1.73	2.64				
IMPHY008983	146 HIS, 146 HIS, 152 HIS	107 THR, 139 THR, 142 HIS, 170 LEU	168 PRO	1.89	2.83	142 HIS	-	−8.51	Tertamine
			170 LEU	3.32	4.05				
CID5362422	UNL, 146 HIS, 146 HIS, 152 HIS	108 LEU, 138 VAL, 139 THR, 142 HIS, 170 LEU,	108 LEU	1.91	2.87	142 HIS	-	−8.0	-
			109 GLY	2.58	3.42				
			143 GLU	1.88	2.72				
			168 PRO	2.1	2.95				
			168 PRO	2.19	2.78				
IMPHY014737	146 HIS, 146 HIS, 152 HIS	107 THR, 170 LEU	108 LEU	2.06	2.97	-	-	−8.54	-
			109 GLY	2.7	3.42				
			167 ARG	2.15	3.02				
			168 PRO	1.89	2.38				
IMPHY008176	146 HIS, 146 HIS, 152 HIS	108 LEU, 138 VAL, 167 ARG, 170 LEU, 176 TYR	168 PRO	3.15	4.09	-	-	−8.13	-
CID132519	146 HIS, 146 HIS, 152 HIS	107 THR, 108 LEU, 110 LEU, 138 VAL, 142 HIS, 170 LEU	105 GLU	3.4	4.07	142 HIS	Aromatic	−8.33	-
			108 LEU	1.9	2.88				
			109 GLY	2.65	3.54				
			139 THR	2.15	3.17				
			168 PRO	2.15	2.99				
			170 LEU	1.64	2.61				
IMPHY003833	UNL, 146 HIS, 146 HIS, 152 HIS	108 LEU, 138 VAL, 139 THR, 142 HIS, 170 LEU	168 PRO	2.42	3.16	142 HIS	-	−7.76	Carboxylate

**Table 2 ijms-25-12675-t002:** DFT-based molecular descriptors of the selected 10 compounds.

S.NO	Compound Name	HOMO (Down)	LUMO (Up)	Band Gap	Chemical Potential	Chemical Hardness	Electro Negativity	Electrophilicity Index	Dipole Moment	Electronic Energy
1	IMPHY011953	−0.18701	−0.0079	0.1792	−0.097435	0.089575	0.097435	0.000425194	1.418062	−1248.58924
2	IMPHY003833	−0.194	−0.0672	0.1268	−0.130585	0.063415	0.130585	0.00054069	5.370888	−1491.6707
3	CID5362422	−0.16827	−0.0767	0.0916	−0.12249	0.04578	0.12249	0.000343437	1.496957	−1210.49432
4	IMPHY001495	−0.20577	−0.0195	0.1863	−0.11263	0.09314	0.11263	0.000590765	0.721364	−1031.35776
5	IMPHY014737	−0.16374	−0.0317	0.132	−0.097735	0.066005	0.097735	0.000315244	2.469465	−1138.91042
6	CID5291	−0.18801	−0.0631	0.125	−0.125535	0.062475	0.125535	0.000492273	5.839185	−1676.63921
7	CID132519	−0.18691	0.007	0.1939	−0.089955	0.096955	0.089955	0.000392275	1.694252	−1601.4156
8	IMPHY008983	−0.16601	−0.0746	0.0914	−0.1203	0.04571	0.1203	0.00033076	5.839185	−1676.63921
9	IMPHY008176	−0.20114	−0.0303	0.1708	−0.11572	0.08542	0.11572	0.000571935	4.669505	−1028.96371
10	IMPHY003823	−0.18595	−0.0475	0.1385	−0.116725	0.069225	0.116725	0.000471586	3.621645	−1106.56823

**Table 3 ijms-25-12675-t003:** Molecular mechanics/Poisson–Boltzmann surface area (MMPBSA) binding energies of Atrolysin and top 10 complexes determined over 20 ns.

S. No.	Compounds	Van der Waal Energy (kJ/mol)	Electrostatic Energy (kJ/mol)	Polar Solvation Energy (kJ/mol)	SASA Energy (kJ/mol)	Binding Energy (kJ/mol)
1	IMPHY011953	−50.053 ± 6.207	−23.340 ± 1.122	44.676 ± 2.568	−7.160 ± 5.030	−35.878 ± 4.410
2	IMPHY003823	−31.576 ± 3.752	−10.865 ± 3.307	33.777 ± 1.526	−4.098 ± 4.608	−12.762 ± 7.796
3	CID5291	−178.487 ± 1.525	3.739 ± 1.125	111.812 ± 1.800	−19.281 ± 1.174	−52.217 ± 17.938
4	IMPHY001495	−108.495 ± 1.472	8.623 ± 11.961	51.790 ± 6.033	−12.936 ± 1.238	−61.017 ± 14.235
5	IMPHY008983	−86.897 ± 1.109	−10.801 ± 0.391	77.989 ± 1.954	−11.525 ± 1.351	−31.235 ± 6.945
6	CID132519	−169.410 ± 4.088	7.300 ± 1.620	105.043 ± 2.657	−17.719 ± 1.342	−74.786 ± 3.357
7	IMPHY014737	−123.130 ± 7.218	−33.572 ± 6.546	120.003 ± 2.766	−13.463 ± 0.722	−50.162 ± 2.842
8	IMPHY008176	−80.811 ± 1.195	−7.925 ± 6.541	47.639 ± 8.642	−10.480 ± 1.212	−51.577 ± 1.677
9	CID5362422	−106.120 ± 1.055	−99.731 ± 6.062	144.271 ± 5.594	−13.195 ± 1.398	−74.775 ± 0.142
10	IMPHY003833	−128.191 ± 8.734	−5.751 ± 8.411	82.401 ± 4.671	−13.535 ± 1.061	−65.076 ± 4.785

## Data Availability

Data are contained within the article and Appendix A.

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
