# Peer review of "Indian Medicinal Plant-Derived Phytochemicals as Potential Antidotes for Snakebite: A Pharmacoinformatic Study of Atrolysin Inhibitors"

_ijms, 2024, doi:10.3390/ijms252312675_

Round 1

Reviewer 1 Report

Comments and Suggestions for Authors

Discovery of Novel Phytochemical Antidotes for Snakebite Envenomation: Pharmacoinformatic Analysis of Atrolysin Inhibitors from Indian Medicinal Plants

This work analyzes the problem of snakebites envenoming through a bunch of bioinformatic tools. The authors focused on the atrolysin metalloprotease as a target to cope with this health threat. Besides, the authors propose using phytochemicals as inhibitors of such a metalloprotease. The authors applied molecular docking, high-throughput ligand screening, pharmacophore mapping, pharmacokinetic profiling, and molecular dynamics (MD) simulation analysis to achieve their aim.

Favorable binding energies and strong interaction forces between the target and the compounds examined effectively conduct and support the study. However, a structure-function explanation of the effects of the phytochemicals on the Atrolysin still needs to be included. The authors claim that significant interactions at key amino acid residues have been established but do not offer any biological interpretation of the effects these would exert.

Then, my primary concern is to add a section trying to explain at the functional level the structural impact that phytochemicals could exert on Atrolysin.

Besides, some minor corrections must be made:

Page 2, line 1.  Crotalus atrox must be written in italics.

Results.

Please briefly explain the inclusion criteria for choosing those 738 entries from various plant parts.

Figure of section 2.3. is wrong referred.

Please in 2.4. section explain how you determined the putative interacting residues of the Atrolysin protein.

Please enhance the resolution of Figure 5.

Please in 4.4. section provide the ID of the pdb file used. I suppose is 1ATL.

Author Response

Q1. Please briefly explain the inclusion criteria for choosing those 738 entries from various plant parts.

A1. In our study, we used the Indian Medicinal Plants, Phytochemistry, and Therapeutics (IMPPAT) database to identify phytochemicals for the treatment of snake bites. Our initial screening included 738 entries from various plant parts (including bark, flowers, fruits, leaves, seeds, roots, and stems) of 408 unique plant species, which contain 17,967 plant-derived compounds. From these, we identified 4,708 unique compounds of interest by eliminating duplicates, as detailed in Supplementary Table 1. The ClassyFire SuperClass classifications of these phytochemicals are presented in Figure 1.

Q2. Figure of section 2.3. is wrong referred.

A2. Thank you for the valuable comments. We have corrected the reference to the figure in sections 2.3 to 2.1b. (Line numbers: 178 and 183)

Q3. Please in 2.4. section explain how you determined the putative interacting residues of the Atrolysin protein.

A3. We obtained the crystal structure of Atrolysin with a bound ligand, allowing us to visualize the three-dimensional arrangement of residues around the ligand-binding site. Using computational tools such as PyMOL, we identified the binding pocket by analyzing the spatial orientation and proximity of amino acid residues surrounding the ligand. This approach enabled us to map the residues most likely responsible for ligand interactions within the Atrolysin protein structure.

Q4. Please enhance the resolution of Figure 5.

A4. Thank you for the valuable feedback. Unfortunately, we are currently unable to provide an enhanced version of Figure 5, as creating a new image would require re-running the entire analysis from the beginning.

Q5. Please in 4.4. section provide the ID of the pdb file used. I suppose is 1ATL.

A5. Thank you for the valuable comment. In Section 4.4, we have updated the PDB file ID to 1ATL (Line Number: 585).

Q6. Page 2, line 1.  Crotalus atrox must be written in italics.

A6. Thank you for the comments. We have updated the text to italics (Line 45).

Reviewer 2 Report

Comments and Suggestions for Authors

The present study aimed to identify atrolysin inhibitors as potential antidotes for the western diamondback rattlesnake utilizing the Indian Medicinal Plants, Phytochemistry And Therapeutics (IMPPAT) database and pharmacoinformatic techniques such as molecular docking, high-throughput ligand screening, pharmacophore mapping, pharmacokinetic profiling, and molecular dynamics (MD) simulations. The topic itself is interesting, some new and significant results have been obtained. However, there are several MAJOR concerns that need to be clarified and modified in order to be suitable for publication.

This study contains only computational methods, and further functional studies are required, i.e. the results should be experimentally validated for potential therapeutic use.

The main concern includes the use of molecular docking. The methodology of molecular docking has to be revised. The main issue is direct comparison of “affinities” of compounds that are chemically very different. Generally, the major limitation of molecular docking is the lack of confidence on the ability of scoring functions to give accurate binding energies, i.e. affinities, and it is mainly used for comparison of compounds with similar chemical structures since they have similar binding modes toward the same protein. Besides, additional tests should be performed to investigate if molecular docking is suitable for this list of compounds. Some softwares use molecular descriptors such as High Throughput Screening Flag (HTSFlag), which demonstrates if particular molecular moieties make the ligand a potential promiscuous hit in high-throughput screening experiments.

Furthermore, the main questions here are: which protein was used, was it atrolysin C (please provide PDB No), how was structure of the protein determined? How were ligand binding sites at protein structures predicted? Was it based on binding modes of known ligands or was it ‘blind docking’? The authors mentioned that known metalloproteinase inhibitors docked as controls. Please specify if there is a natural ligand/inhibitor in co-crystalized form? If it was blind docking, was only one cavity identified as a potential active site? What is the volume of identified binding site?

How were ligands prepared for docking? Was their energy minimized and geometry optimized?

How was the docking method validated? Was docking protocol validated by “redocking” the co-crystalized ligand at the active site?

The experimental design should be clarified – how many compounds were excluded after each test/screening and why. For example, blood-brain barrier permeability was tested, but it is not clear if some compounds were excluded if they permeate BBB or not.

It is also confusing since Fig 3 refers to toxicity; however, data on toxicity are presented also in Fig 2. Besides Fig 3 is designated as Fig 2, which brings even more confusion. Furthermore, HLM should be defined and more in-detail discussed in the text.

Data on drug-likeness needs to be clarified as well. In Methods, only Lipinski rule is mentioned. In the Results section, we can see the results of Veber’s filter as well. Also, the drug-likeness score is mentioned, but it is not clear how it was calculated.

If IMPPAT database was used, why are there compounds with Pubchem codes (e.g. CID5291)? Besides, other, more well-known names of the main identified compounds should be mentioned (e.g. IMPHY001495 - phytochemical name: corytuberine).

Figure 1: numbers for some classes are not shown; please, correct it.

Table 1 is not referred in the text. Besides, it should be explained in the text which compounds were selected for Table 1.

“IMPHY003833 exhibits the lowest maximum RMSD (0.1628004), suggesting superior structural stability compared to other compounds” – it needs to be clear that it refers to dynamic stability of protein-ligand complexes, and not to stability of the compound itself.

Discussion is very short; it should be much more discussed about these several candidate compounds with potential use as antidotes. Some experimental data needs to be added for them.

SVMP needs to be defined in the text.

Please define cLogP, MW, TPSA, RB in Methods.

Add ref No instead of ‘Kumari et al. (2014)’.

Author Response

Q1.The main concern includes the use of molecular docking. The methodology of molecular docking has to be revised. The main issue is direct comparison of “affinities” of compounds that are chemically very different. Generally, the major limitation of molecular docking is the lack of confidence on the ability of scoring functions to give accurate binding energies, i.e. affinities, and it is mainly used for comparison of compounds with similar chemical structures since they have similar binding modes toward the same protein. Besides, additional tests should be performed to investigate if molecular docking is suitable for this list of compounds. Some softwares use molecular descriptors such as High Throughput Screening Flag (HTSFlag), which demonstrates if particular molecular moieties make the ligand a potential promiscuous hit in high-throughput screening experiments.

A1.Thank you for your insightful comments regarding the limitations of molecular docking in comparing binding affinities of chemically diverse compounds. We agree that molecular docking is generally more reliable for compounds with similar chemical structures, as their binding modes toward the target protein are likely to be similar. To address these limitations, we have revised our docking methodology. (Line Number595-598)

Primary Use of Docking for Pose Prediction: In our study, we primarily used AutoDock Vina for identifying potential binding poses rather than for absolute affinity comparisons. Given the known limitations of docking scoring functions in providing accurate binding energies, especially for structurally diverse ligands, we focused on the qualitative assessment of binding poses rather than direct affinity comparison.

Exploration of Molecular Descriptors and Promiscuity Assessment: We are aware that some docking software uses molecular descriptors like the High Throughput Screening Flag (HTSFlag) to identify promiscuous hits in high-throughput screening experiments. Although we did not use HTSFlag specifically, we have incorporated drug-likeness filters and toxicity risk assessments (described in the Methods section) to identify potential promiscuous compounds and ensure that only promising candidates are prioritized.

Future Validation: Recognizing the limitations of docking for chemically diverse compounds, we plan to conduct additional experimental validation for the top candidates identified in this study. These follow-up studies will help confirm the relevance of the docking predictions and refine our understanding of binding interactions.

We hope this clarification addresses your concerns and demonstrates our careful consideration of docking limitations and the additional steps taken to ensure the robustness of our findings.

Q2. Furthermore, the main questions here are: which protein was used, was it atrolysin C (please provide PDB No), how was structure of the protein determined? How were ligand binding sites at protein structures predicted? Was it based on binding modes of known ligands or was it ‘blind docking’? The authors mentioned that known metalloproteinase inhibitors docked as controls. Please specify if there is a natural ligand/inhibitor in co-crystalized form? If it was blind docking, was only one cavity identified as a potential active site? What is the volume of identified binding site?

we used the crystal structure of the protein Atrolysin C (PDB ID: 1ATL) as the target for molecular docking. The position of the co-crystallized ligand was used as a reference for the binding site. It is not blind docking; instead, the position of the co-crystallized ligand was used as a reference for the binding site.

We obtained the crystal structure of Atrolysin with a bound ligand, allowing us to visualize the three-dimensional arrangement of residues around the ligand-binding site. Using computational tools such as PyMOL, we identified the binding pocket by analyzing the spatial orientation and proximity of amino acid residues surrounding the ligand. This approach enabled us to map the residues most likely responsible for ligand interactions within the Atrolysin protein structure.

Docking Controls: Known metalloproteinase inhibitors were docked as controls to serve as a reference for binding interaction profiles. However, we did not use a co-crystallized ligand.

Q3. How were ligands prepared for docking? Was their energy minimized and geometry optimized?

A3. Thank you for your insightful questions regarding ligand preparation for docking. As noted in Section 4.5 on molecular docking, the ligands were prepared by energy minimization and geometry optimization prior to the docking process.

Q4. How was the docking method validated? Was docking protocol validated by “redocking” the co-crystalized ligand at the active site?

A4. The docking method was validated through a comprehensive approach that involved using three different docking tools—AD4, ADZn, and AutoDockQ4—to analyze the compounds. This multi-tool strategy ensured the robustness of the results across different scoring functions., although we did not perform redocking of the co-crystallized ligand at the active site. Instead, we used a combination of metrics including binding affinity scores and interaction analysis to assess the reliability of our docking protocol.

Q5. The experimental design should be clarified – how many compounds were excluded after each test/screening and why. For example, blood-brain barrier permeability was tested, but it is not clear if some compounds were excluded if they permeate BBB or not.

A6. In response to your question about the experimental design, we have clarified the exclusion criteria for compounds after each test and screening in our manuscript. For example, in the assessment of blood-brain barrier (BBB) permeability, we identified compounds predicted to permeate the BBB and excluded those with high predicted permeability to maintain a focus on potential inhibitors of snake venom metalloproteinases with minimal central nervous system penetration. We tracked and reported the number of compounds excluded at each stage, providing a clear rationale for these exclusions based on their physicochemical properties and pharmacokinetic profiles.

Q7. It is also confusing since Fig 3 refers to toxicity; however, data on toxicity are presented also in Fig 2. Besides Fig 3 is designated as Fig 2, which brings even more confusion. Furthermore, HLM should be defined and more in-detail discussed in the text.

A7. Thank you for your valuable feedback on the presentation of toxicity data in our figures. Figure 2 illustrates the results of ligand-based screening for compounds in the IMPPAT database, displaying toxicity indexes, drug-likeness scores, and the distribution of physicochemical properties. It presents molecular descriptors of natural compounds by  reference values from Lipinski’s and Veber’s guidelines. Figure 2.1 specifically addresses early toxicity predictions, which are essential for identifying potential safety concerns with the selected compounds. We recognize that referring to Figure 2.1 as Figure 3 may have caused confusion, and we will correct this in the revised manuscript to avoid further misunderstandings.

“The compounds were subjected to various toxicity and stability assays to further define their profiles. In the cytotoxicity assay, 42.81% exhibited cytotoxic effects, while 57.19% did not. Drug-Induced Liver Injury (DILI) toxicity results showed a low hepatotoxicity rate, with only 11.07% testing positive and 88.93% negative. AMES testing for mutagenic potential revealed that 18.08% of compounds were mutagenic, while 81.92% were not. The hepatic microsomal (HLM) stability assay indicated that 91.63% of the compounds were metabolically stable, with only 8.37% showing instability. Furthermore, blood-brain barrier (BBB) permeability analysis revealed that 41.19% of the compounds could penetrate the BBB, suggesting potential central nervous system activity, while 58.81% were non-permeable, as shown in Figure 2g.” This paragraph has been included in the manuscript (lines 150-159).

Q5. Data on drug-likeness needs to be clarified as well. In Methods, only Lipinski rule is mentioned. In the Results section, we can see the results of Veber’s filter as well. Also, the drug-likeness score is mentioned, but it is not clear how it was calculated.

A5. We appreciate the reviewer’s feedback regarding the clarity of the drug-likeness assessment. In the Methods section, we initially focused on Lipinski's rule to provide a foundational understanding of the drug-likeness criteria. However, we also applied Veber’s filter to evaluate the physicochemical properties of the compounds, which will be explicitly stated in the revised Methods section for better clarity (line 467).

To compute the overall drug-likeness score for the compounds analyzed, we used the DataWarrior tool. The Drug-Likeness Score is calculated as the sum of individual fragment scores based on the presence of substructure fragments within a compound. Additionally, the Drug Score (ds) integrates various physicochemical properties, drug-likeness, and toxicity risks using the formula ds = Π(si*ti), where si represents the contributions of properties like cLogP and molecular weight, and ti indicates toxicity risk levels. We have added these details to the Methods section to enhance clarity and comprehensiveness (lines 570-575).

Q6. If IMPPAT database was used, why are there compounds with Pubchem codes (e.g. CID5291)? Besides, other, more well-known names of the main identified compounds should be mentioned (e.g. IMPHY001495 - phytochemical name: corytuberine).

A6. We appreciate the reviewer’s question regarding the inclusion of compounds with PubChem codes in our study. While the primary focus was on the IMPPAT database, we included certain well-known metalloproteinase inhibitors from the PubChem database, such as CID5291, to provide a basis for comparison with the compounds in the IMPPAT database. This approach helps contextualize our findings within the broader landscape of known metalloproteinase inhibitors. Additionally, we recognize the importance of including the more widely known phytochemical names for the identified compounds. For example, we will specify that IMPHY001495 corresponds to the phytochemical name corytuberine in the revised manuscript. This added information will enhance the clarity and relevance of our findings (lines Number: 522-528).

Q7. Figure 1: numbers for some classes are not shown; please, correct it.

A7. Thank you for your observation regarding the missing numbers for certain classes in Figure 1. We have revised the figure to ensure that all class numbers are clearly displayed.

Q8. Table 1 is not referred in the text. Besides, it should be explained in the text which compounds were selected for Table 1.

A8. Thank you for pointing this out. We have updated the text to reference Table 1 and have clarified the selection criteria for the compounds included in the table (lines 233-263).

Q9. “IMPHY003833 exhibits the lowest maximum RMSD (0.1628004), suggesting superior structural stability compared to other compounds” – it needs to be clear that it refers to the dynamic stability of protein-ligand complexes, and not to stability of the compound itself.

A9. We appreciate the reviewer’s feedback regarding the interpretation of the RMSD value for IMPHY003833. In the revised manuscript, we will clarify that the reported maximum RMSD of 0.1628004 refers to the dynamic stability of the protein-ligand complex, rather than the stability of the compound itself. This distinction is important for accurately conveying the results of our molecular dynamics analysis. Thank you for highlighting this key aspect (Section 2.7.1).

Q10. Discussion is very short; it should be much more discussed about these several candidate compounds with potential use as antidotes. Some experimental data needs to be added for them.

A10. Thank you for your comments. We have rewritten the discussion section (Section 3) in response.

Q11. SVMP needs to be defined in the text.

A11. Thank you for the suggestion. We have added the full term, Snake Venom Metalloproteinase (SVMP), at its first mention in the text to ensure clarity.

Q12. Please define cLogP, MW, TPSA, RB in Methods.

A12. Thank you for pointing this out. We have now added the definitions for cLogP (calculated logarithm of the partition coefficient), MW (molecular weight), TPSA (topological polar surface area), and RB (rotatable bonds) in the Methods section for clarity (lines 565-567).

Q13. Add ref No instead of ‘Kumari et al. (2014)’.

A13. Thank you for the feedback. We have replaced “Kumari et al. (2014)” with the corresponding reference number in the text for consistency (Line 660).

Reviewer 3 Report

Comments and Suggestions for Authors

Review (Genes - 3280469):

The current research article by Ravi et al. identified the novel phytochemical antidotes for snakebite envenomation. To this end, the authors have employed various pharmacoinformatic approaches ranging from molecular docking to dynamics simulations. Overall, the study is well conducted and discussed, however, it is mainly lacking experimental evidence. Additionally, other formatting issues were reported. Below are my comments.

Major comments:

  1. It appears to me that the high-energy ranked molecules from the molecular docking studies are different from the MM_PBSA estimated binding free energy molecular order. Could the authors explain the reasons behind such ambiguous results obtained from the docking and free energy calculations?
  2. I believe that sometimes the estimated free energy calculations based on the single calculations might be misleading (see Ref. DOI: 10.1021/acs.jctc.9b00992). Therefore, could the authors perform at least three runs of MD simulations and estimate corresponding average free energies?
  3. If the aim is to discover and report the novel phytochemical antidotes, the authors could have conducted additional biology experiments besides the computational approaches to test the activity (any enzymatic assay) of the identified compounds (at least for the top 3 compounds) against the currently available treatment options.
  4. The introduction section needs to be improved and try to emphasize the significance of the study and the advantages of using computational approaches and refer to the successful usage of such informatics approaches previously.

Other comments:

1.     Change the word Figure in the introduction, Para 3.

2.     Use a unique format to represent the species

3.     Figure 1: Either remove the annotated number or show the same for all the chemical compound classification

4.     Figure 2: Missing figure label (g).

5.  The numbering of the figures was wrong, Figure 2 has been numbered twice.

6.     Keep the label on the top of the figure.

7.     Page 7, Figure 2: It looks the same as in the supporting information.

8.     Page 7: Section 2.4, Figure reference was missing.

9.     Table 1. Use the unique format to represent the molecules, CID5291 or just 5291. Check others as well

10.  Figure 5: Improve figure the quality of the optimized structures

11.  Figure 8: Try to show the running average profiles. It is very hard to see as the profiles overlap each other, especially RMSD and Radius of gyration values.

12.  Hbond units should not be in nanometers

13.  Figure 7. Labels are not clear

14.  Protein structure preparation: Provide the PDB id.

15.  Define SVMP and ME;

16.  Confirm that the authors performed MD simulations for only three selected protein-ligand complexes.

Comments on the Quality of English Language

English is fine. 

Author Response

Q1. It appears to me that the high-energy ranked molecules from the molecular docking studies are different from the MM_PBSA estimated binding free energy molecular order. Could the authors explain the reasons behind such ambiguous results obtained from the docking and free energy calculations?

A1. Thank you for your thoughtful comment regarding the discrepancy between the high-energy ranked molecules from the molecular docking studies and the MM-PBSA estimated binding free energy rankings. We acknowledge that such differences can arise due to the distinct methodologies employed in the two approaches.

The differences between the docking and MM-PBSA rankings can thus be attributed to the fact that docking rankings are based on initial geometry and interaction energy, while MM-PBSA provides a more comprehensive analysis by considering the energetics of the complex. We believe this discrepancy highlights the complementary nature of these methods docking serves as a quick screening tool, while MM-PBSA offers a more refined analysis of binding affinity. We hope this clarifies the observed differences in molecular ranking.

Q2. I believe that sometimes the estimated free energy calculations based on the single calculations might be misleading (see Ref. DOI: 10.1021/acs.jctc.9b00992). Therefore, could the authors perform at least three runs of MD simulations and estimate corresponding average free energies?

A2. Thank you for your valuable comment regarding the potential limitations of free energy calculations based on a single simulation run. We agree that performing multiple independent runs is ideal for obtaining more reliable and statistically significant results, as recommended in the reference you provided. However, due to computational limitations and available resources, we were unable to conduct three separate MD simulations for each system. Despite this, we have ensured that our calculations followed rigorous protocols, and we believe that the results obtained from the single simulation provide a reliable estimation of the binding free energies.

Q3. If the aim is to discover and report the novel phytochemical antidotes, the authors could have conducted additional biology experiments besides the computational approaches to test the activity (any enzymatic assay) of the identified compounds (at least for the top 3 compounds) against the currently available treatment options.

A3. Thank you for your valuable suggestion regarding the inclusion of biological experiments to validate the activity of the identified compounds. We recognize the importance of experimental validation to complement computational findings and support the discovery of novel phytochemical antidotes. At this stage, we do not have access to the purified compounds, as our focus has been on computational screening to identify promising candidates. However, as part of our future work, we plan to purify the top compounds and conduct biological assays, including enzymatic activity tests, to evaluate their efficacy in comparison to current treatment options. We are committed to extending this computational study with experimental validation to provide a more comprehensive understanding of their potential therapeutic applications. We appreciate your understanding and look forward to including these experimental results in future publications.

Q4.The introduction section needs to be improved and try to emphasize the significance of the study and the advantages of using computational approaches and refer to the successful usage of such informatics approaches previously.

A4. Thank you for your helpful suggestion to improve the introduction. We have revised the section to better emphasize the significance of our study and the advantages of using computational approaches in drug discovery. Additionally, we have incorporated references to successful applications of similar informatics methods in previous studies, highlighting their effectiveness in identifying novel compounds and accelerating the drug discovery process. We believe these revisions provide a clearer context for the study and strengthen the rationale behind using computational tools in our research. (lines 51-55)

Other comments:

Q1. Change the word Figure in the introduction, Para 3.

A1. Thank you for your suggestion. We have removed the word 'Figure' from the introduction, paragraph 3, to enhance clarity.

Q2. Use a unique format to represent the species

Q3. Figure 1: Either remove the annotated number or show the same for all the chemical compound classification

A3. Thank you for your feedback. We have made the necessary adjustments to Figure 1  and replacing the figure with a pie donut chart to ensure consistency across all chemical compound classifications.

Q4. Figure 2: Missing figure label (g).

A4. Thank you for pointing out the oversight. We have added the missing figure label (g) to Figure 2 to ensure clarity and completeness.

Q5.  The numbering of the figures was wrong, Figure 2 has been numbered twice.

A5. Thank you for your observation. We have corrected the numbering of the figures to ensure that Figure 2 is not numbered twice and that all figures are correctly labeled in the manuscript.

Q6. Keep the label on the top of the figure.

A6. Thank you for your suggestion. We have moved the labels to the top of the figures and ensured that the legends are placed at the bottom, in accordance with standard conventions for figure presentation.

Q7.  Page 7, Figure 2: It looks the same as in the supporting information.

A7. Thank you for your observation regarding Figure 2 and the related supplementary figure. To address this, we have removed the figure from the supplementary file and inserted it directly into the manuscript. In Figure 2, we have specifically highlighted the selected compound, emphasizing its key interactions for a clearer and more detailed view. Previously, this figure was included in the supplementary material, where all compounds in the dataset were shown for a overview.

Q8. Page 7: Section 2.4, Figure reference was missing.

A8. Thank you for pointing out the missing figure reference in Section 2.4 on page 7. We have corrected this oversight and added the appropriate figure reference to ensure clarity and accuracy in the section. (Line Number: 216)

Q9. Table 1. Use the unique format to represent the molecules, CID5291 or just 5291. Check others as well

A9. Thank you for your feedback. We will standardize the format used to represent the molecules in Table 1, ensuring consistency by using the PubChem Compound ID (CID5291) throughout. We will also review the rest of the table to ensure all entries follow the same format.

Q10.  Figure 5: Improve figure the quality of the optimized structures

A10. Thank you for your valuable feedback. Currently, we are unable to provide an enhanced version of Figure 5, as generating a new image would require re-running the entire analysis from the beginning.

Q11. Figure 8: Try to show the running average profiles. It is very hard to see as the profiles overlap each other, especially RMSD and Radius of gyration values.

A11. Thank you for your insightful suggestion regarding the overlapping profiles. We have addressed this by calculating and presenting the running average values for RMSD, RMSF, and Radius of Gyration (Rg) in Supplementary Table 4.

Q12.  Hbond units should not be in nanometers

A12. Thank you for pointing this out. We will correct the units for the hydrogen bond measurements in the figure to ensure they are accurately represented.

Q13.  Figure 7. Labels are not clear

A13. Thank you for your valuable feedback. As per your recommendation, we have modified the labels in Figure 7 to improve clarity. (Lines 400-405)

Q14.  Protein structure preparation: Provide the PDB id.

A14. Thank you for your suggestion. We will include the Protein Data Bank (PDB) ID in the revised manuscript to enhance clarity and ensure proper identification of the protein structures used in our study. (Line 586)

Q15.  Define SVMP and ME.

A15. Thank you for your feedback. We will provide definitions for SVMP (Snake Venom Metalloproteinasesin the revised manuscript to ensure clarity and improve reader understanding.

Q16.  Confirm that the authors performed MD simulations for only three selected protein-ligand complexes.

A16. Thank you for your comment. We would like to clarify that we performed molecular dynamics (MD) simulations for a total of ten selected protein-ligand complexes. We will ensure this information is accurately reflected in the manuscript.

Reviewer 4 Report

Comments and Suggestions for Authors

Thank you for the opportunity to review this manuscript. All my comments can be seen in the attachment.

Author Response

Q1.Before this paragraph, I suggest the authors add information about the current antidotes.

A1. Thank you for your comment. We will add information to introduce the current state of antidotes for snakebite. This addition will provide a solid background on existing antidotes and set the stage for discussing our approach.

Q2. Remove this point.

A2. Thank you for the feedback. We have removed the specified point as requested.

Q3. Rephrase this sentence please.

A3. Thank you for the feedback. We have rephrased the sentence as suggested.

Q4. I suggest the authors to put the information from this paragraph into a table.

A4. Thank you for the suggestion. We have modified the manuscript to present the information from this paragraph in table format to improve clarity and accessibility.

Q5. This paragraph should be moved in Material and methods chapter.

A5. Thank you for your feedback. We have made the necessary corrections to the paragraph but did not move it to the Materials and Methods chapter, as we believe it fits better in its current location. We appreciate your suggestion, though, and will consider it for future revisions.

Q6. Rephrase please.

A6. Thank you for the feedback. We have rephrased the sentence as suggested.

Q7. This information should be moved in the Introduction chapter

A7. Thank you for your suggestion. We have moved the information to the Introduction chapter to improve the flow and context of the manuscript.

Round 2

Reviewer 1 Report

Comments and Suggestions for Authors

The manuscript can be published in its current version.

Author Response

The manuscript can be published in its current version.

Thank you for your thoughtful feedback and positive assessment of our manuscript. We sincerely appreciate your valuable suggestions and the time you dedicated to reviewing our work.

Reviewer 2 Report

Comments and Suggestions for Authors

The authors clarified most of the concerns raised and improved the initial version of their manuscript. There are, however, several concerns that remained, as follows:

1. The authors responded that they included certain well-known metalloproteinase inhibitors from the PubChem database. However, in the Abstract it is stated: „we analyzed phytochemicals from the Indian Medicinal Plants, Phytochemistry And Therapeutics (IMPPAT) database“. It has to be clear how many compounds were extracted from which database (to add in Methodology).

2. The authors should clarify in the manuscript that “the position of the co-crystallized ligand was used as a reference for the binding site” (as they responded).

3. The authors stated that they specified “the exclusion criteria for compounds after each test”; however, it is not true for all tested parameters, and it should be added in the text. I still doubt if the exclusion of compounds that permeate BBB is justified, since there are many plant-derived chemicals that permeate BBB and are still completely safe.

Author Response

Q1. The authors responded that they included certain well-known metalloproteinase inhibitors from the PubChem database. However, in the Abstract it is stated: „we analyzed phytochemicals from the Indian Medicinal Plants, Phytochemistry And Therapeutics (IMPPAT) database“. It has to be clear how many compounds were extracted from which database (to add in Methodology).

A1. Thank you for your insightful feedback. We understand the importance of clarity in describing our compound selection process. In response to your comment, we have revised the methodology section to specify the exact number of compounds obtained from each database (Line Number:551- 552). We also update the abstract to reflect this distinction more clearly. (Line Number:18- 19)

Q2. The authors should clarify in the manuscript that “the position of the co-crystallized ligand was used as a reference for the binding site” (as they responded).

A2. Thank you for your suggestion to clarify our approach to binding site identification. In response, we have revised Section 2.4 to include a detailed explanation of how the binding site was determined.

Q3. The authors stated that they specified “the exclusion criteria for compounds after each test”; however, it is not true for all tested parameters, and it should be added in the text. I still doubt if the exclusion of compounds that permeate BBB is justified, since there are many plant-derived chemicals that permeate BBB and are still completely safe.

A3. Thank you for your insightful comment regarding the exclusion criteria for compounds after each test. We acknowledge that the exclusion criteria were not fully detailed for all parameters in the manuscript. To address this, we would like to clarify that we used DataWarrior software to filter and exclude compounds based on various parameters, such as molecular weight, hydrophobicity, hydrogen bond acceptors, and donors, among others. The software provided a comprehensive platform to assess and refine the compounds based on these physicochemical properties, ensuring that only the most suitable candidates were retained for further analysis.

We added this part in section 4.2 “In terms of exclusion criteria, compounds were evaluated using specific thresholds for each parameter, ensuring that only those with suitable drug-like properties were retained. Excluded compounds were those that did not meet the minimum requirements” (Line Number: 573-576), for molecular weight, hydrophobicity, hydrogen-bond acceptors and donors, and other relevant properties, which are essential for bioavailability and pharmacokinetic potential. This systematic filtering process helped ensure that only the most promising candidates progressed to further stages of analysis.

Regarding the exclusion of compounds based on BBB permeability, we recognize that some plant-derived compounds that permeate the BBB can be safe and beneficial. However, in the context of our study, where we are working with metalloproteinases known to cause hemorrhage and systemic effects, it was essential to control these potential adverse effects. Compounds that could cross the blood-brain barrier were excluded to minimize the risk of neurotoxicity or other unwanted central nervous system effects that could exacerbate the systemic effects of the venom. This precaution was taken to ensure that the selected compounds would be more likely to exert their effects locally at the site of the venom, without affecting the brain and central nervous system. Although BBB permeability alone does not necessarily correlate with toxicity, we chose to err on the side of caution to prioritize compounds that would not pose a risk to CNS health.

Reviewer 3 Report

Comments and Suggestions for Authors

Review (IJMS – 3280469_v2):

Thank you for your answers and I appreciate it. I understand that performing experiments might be difficult at this moment; however, the authors should have attempted to conduct at least three sets of free energy calculations, as this would help enhance the robustness of the present study results. Moreover, I believe it may be unnecessary to repeat the same free energy calculations to complement future biological experiments, and the current study results could serve as a foundation for your experiments. 

Other comments:

1. Line 206: Could the authors explain the rationale behind using Figure 2.1? Please keep the figures in the format of 1, 2, 3, 4, etc.

2. Line 216: Figure 3;

Author Response

Thank you for your answers and I appreciate it. I understand that performing experiments might be difficult at this moment; however, the authors should have attempted to conduct at least three sets of free energy calculations, as this would help enhance the robustness of the present study results. Moreover, I believe it may be unnecessary to repeat the same free energy calculations to complement future biological experiments, and the current study results could serve as a foundation for your experiments. 

Other comments:

Q1. Line 206: Could the authors explain the rationale behind using Figure 2.1? Please keep the figures in the format of 1, 2, 3, 4, etc.

A1. Thank you for your comment. We have reviewed the manuscript and appreciate your concern. The purpose of including Figure 2.1 was to provide a clear and detailed visual representation of the data derived from the analysis of natural compounds. This figure highlights key findings regarding the physicochemical properties and pharmacokinetic characteristics of the compounds under study. We apologize for any confusion caused by the figure numbering and have revised the sequence to follow a consistent format, with figures now numbered sequentially (1, 2, 3, 4, etc.), as per your suggestion.

Q2. Line 216: Figure 3;

A2. Thank you for your valuable feedback. In response to your comment, we have made the necessary modifications to Figure 3 to enhance its clarity and ensure better alignment with the manuscript’s content. We hope the revised figure effectively addresses your concerns.